# Video-Enhanced Offline Reinforcement Learning: A Model-Based Approach

**Minting Pan** [1]  **Yitao Zheng** [1]  **Jiajian Li** [1]  **Yunbo Wang** [1]  **Xiaokang Yang** [1]

## Abstract

Offline reinforcement learning (RL) enables policy optimization using static datasets, avoiding the risks and costs of extensive real-world exploration. However, it struggles with suboptimal offline behaviors and inaccurate value estimation due to the lack of environmental interaction. We present **Video-Enhanced Offline RL (VeoRL)**, a model-based method that constructs an interactive world model from diverse, unlabeled video data readily available online. Leveraging model-based behavior guidance, our approach transfers commonsense knowledge of control policy and physical dynamics from natural videos to the RL agent within the target domain. VeoRL achieves substantial performance gains (over $100\%$ in some cases) across visual control tasks in robotic manipulation, autonomous driving, and open-world video games. Project page: https://panmt.github.io/VeoRL.github.io.

## 1. Introduction

When humans tackle long-term control tasks they are unfamiliar with, they typically start by watching instructional videos on platforms like YouTube to develop an abstract understanding of the task, before transitioning to hands-on practice in a real environment to refine their skills. Can machines replicate this learning process? In this work, we introduce video-enhanced offline reinforcement learning (VeoRL) to explore this problem.

Offline reinforcement learning refers to the scenario where an RL system is restricted to learning from a limited, static dataset. This approach offers several benefits, as it allows agents to learn from existing data, reducing the need for costly and potentially unsafe real-world engagement (Fujimoto et al., 2019; Kumar et al., 2019; 2020; Levine et al.,

2020; Kostrikov et al., 2022). However, a key challenge in offline RL lies in the inherent incompleteness of fixed datasets, which can result in distributional shifts from the target environment. This issue, compounded by the well-documented overestimation bias in value approximations, creates significant difficulties for current offline RL methods (Yu et al., 2022; Zang et al., 2023; Lu et al., 2023).

The core insight of VeoRL is to leverage unlabeled Internet videos to broaden the agent's understanding of human behaviors and the real physical world, going beyond the limited scope of fixed, predefined datasets. Notably, the vast volume of online video data possesses several key advantages: (i) It can be acquired at a significantly lower cost compared to data collected through real interactions with the target environments. (ii) It implicitly contains demonstration policies that, while potentially suboptimal, can still be beneficial for current visual control tasks.

Nonetheless, the online videos typically lack action labels or reward function annotations and may present remarkable data distributional shifts compared to the target RL domain. These properties introduce unique challenges when using the auxiliary videos for policy optimization. Existing RL pretraining methods generally operate within the raw action space (*e.g.,* through behavior cloning) or perform action-free model warmup (*e.g.,* R3M (Nair et al., 2022) and APV (Seo et al., 2022)), making them ill-suited for transferring underlying behaviors from unlabeled videos.

To address these challenges, we begin by constructing a discrete latent action space to recover the underlying, unobservable control policies from raw video data (see Figure 1a). Specifically, we model the temporal events in the video as a Markov decision process with a continuous state space and a discrete action space. To optimize a finite set of "*behavior abstractions*", we apply vector quantization techniques and use the learned behavior vectors to drive the training of a video prediction model. Empirical results demonstrate that the latent behaviors learned through this approach exhibit strong semantic alignment with the actual actions observed in the target environment.

VeoRL is essentially a model-based RL algorithm, where the agent interacts with a world model that generates state transitions and reward feedback. We train a world model comprising two state transition branches: One branch pre-

[1]MoE Key Lab of Artificial Intelligence, AI Institute, Shanghai Jiao Tong University. Correspondence to: Yunbo Wang <yunbow@sjtu.edu.cn>.

*Proceedings of the 42nd International Conference on Machine Learning*, Vancouver, Canada. PMLR 267, 2025. Copyright 2025 by the author(s).

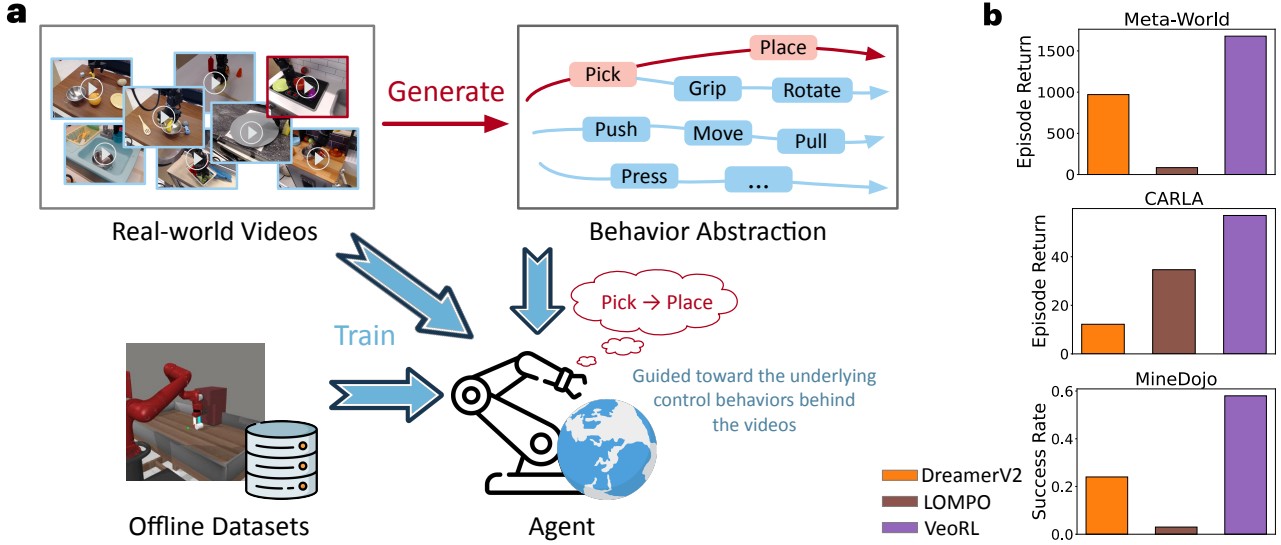

Figure 1. **Overview of the training setup of VeoRL that leads to improved offline RL performance. a**, We extract latent behavior abstractions from task-agnostic, unannotated natural video data to enrich the world model's commonsense understanding of the physical world. By interacting with this world model, the agent performs policy optimization guided explicitly by the latent policies learned from the natural videos. **b**, Overall performance of VeoRL. It demonstrates significant improvements over existing offline RL methods across different visual control benchmarks, including Meta-World robotic manipulation, CARLA autonomous driving, and Minecraft open-world video games. We present the average performance on all tasks on each benchmark.

dicts future state evolution based on the agent's real actions (*trunk net*), while the other branch predicts longer-term environmental feedback derived from latent behaviors (*plan net*). We further introduce an intrinsic reward that encourages the state-action rollouts along the trunk net to progressively align with the long-term future state predicted by the plan net. This approach connects the target policy with the behavior abstractions learned from auxiliary videos, while mitigating the overestimation bias that arises from solely relying on environmental rewards to train the value network.

We evaluate VeoRL on a diverse set of visual control benchmarks. For the Meta-World robotic manipulation tasks, we use the BridgeData-V2 dataset (Walke et al., 2023) as the source of auxiliary real-world videos. For the CARLA autonomous driving tasks, we employ the NuScenes dataset (Caesar et al., 2019), a collection of 1,000 diverse real-world driving scenes, as the auxiliary source domain. For the MineDojo open-world gaming environment, where the agent must navigate a vast state space, we use online Minecraft videos created by human players to provide unlabeled video demonstrations. As summarized in Figure 1b, VeoRL achieves substantial performance improvements over existing RL approaches.

## 2. Related Work

**Model-based visual RL.** For visual control tasks, agents learn control policies from high-dimensional visual inputs. By explicitly modeling state transitions and predicting be-

havioral outcomes, model-based RL approaches (Ha & Schmidhuber, 2018; Hafner et al., 2019; 2020) typically offer better sample efficiency than model-free methods (Yarats et al., 2021b; Kostrikov et al., 2021; Laskin et al., 2020). World Models (Ha & Schmidhuber, 2018) were introduced to first learn compressed latent representations of the environment in a self-supervised manner, and then use these latent states to train the agent. Dreamer (Hafner et al., 2020) and DreamerV2 (Hafner et al., 2021) optimize behavior by predicting and then acting upon expected values over the latent states generated by the Recurrent State-Space Model. Recently, research on goal-conditioned RL using hierarchical world model architectures (Veeriah et al., 2021; Rao et al., 2022; Hafner et al., 2022; Gumbsch et al., 2023; Park et al., 2024) has shown how decomposing tasks into subgoals can facilitate efficient decision-making in complex environments. However, none of these methods address the challenges of offline RL or facilitate unsupervised policy transfer from enriched out-of-domain video data.

**Offline RL.** Offline RL offers a cost-effective alternative by reducing interactions with online environments. Previous research has primarily focused on addressing the challenge of value overestimation, falling into two broad categories: model-free methods (Cen et al., 2024; Wu et al., 2019; Kumar et al., 2020; Osa & Harada, 2024; Li et al., 2023; Fujimoto & Gu, 2021; Hong et al., 2024) and model-based approaches (Chen et al., 2023; Yu et al., 2021; Hatch et al., 2023; Mazoure et al., 2023; Hatch et al., 2022; Sun et al., 2023; Rigter et al., 2022; Yang et al., 2024b). For example,

CQL (Kumar et al., 2020) introduces a model-free offline RL method that mitigates the value overestimation bias by learning a conservative state-action value function. This method can be seamlessly integrated with existing visual RL approaches. In the context of offline visual RL, an additional challenge is effectively handling high-dimensional visual inputs while preventing overfitting during the representation learning of latent states (Shah et al., 2022; Tian et al., 2020; Yadav et al., 2023; Nair et al., 2022; Wang et al., 2024). Specifically, LOMPO (Rafailov et al., 2021) presents a model-based offline visual RL approach that explicitly manages the dynamics uncertainty by incorporating a reward penalty during world model training. In this work, we compare our approach with the model-based LOMPO method and the model-free CQL method integrated into the visual RL backbone of DrQ-V2 (Yarats et al., 2021a).

**Video-enhanced RL.** In recent years, there has been growing interest in harnessing auxiliary video data to enhance RL, as such data is more accessible and affordable in real-world settings. We can broadly categorize these approaches into two groups. The first group focuses on using videos to improve agent performance through representation learning (Seo et al., 2022; Ma et al., 2023; Wu et al., 2023). For instance, APV (Seo et al., 2022) pre-trains an *action-free* video prediction model using videos and then finetunes the agent with these pretrained representations. VIP (Ma et al., 2023) enhances generalization to downstream offline tasks by leveraging self-supervised visual representations. The second group focuses on learning transferable behavior policies from the source video domain (Torabi et al., 2018; Chang et al., 2022; Baker et al., 2022; Schmeckpeper et al., 2021; Ye et al., 2022). For example, the VPT model (Baker et al., 2022) trains an inverse dynamics model with a small set of *labeled* auxiliary videos and then assigns pseudo-actions to unlabeled video data. This method requires strong consistency across the source to target domains. Another line of work focuses on learning interactive world models from the unlabeled *target-domain* videos to generate playable virtual environments (Valevski et al., 2024; Yang et al., 2024a). While promising, these methods highlight the potential of interactive video generation, without explicitly using them for downstream control tasks. Besides, these approaches have yet to be formally published. In summary, unlike VeoRL, previous video-enhanced approaches typically require the source video set to be either annotated with action/reward labels or derived from the same domain as the target environment for policy deployment, or not directly designed for sequential decision-making tasks.

## 3. Method

We aim to enhance the performance of offline visual RL trained on an offline target dataset $\mathcal{B}^{\text{tar}}$ by leveraging task-agnostic video data from a source domain $\mathcal{B}^{\text{src}}$. Two key

challenges arise in this context: (i) The auxiliary videos generally lack action labels and reward function annotations; (ii) There exist inherent distributional shifts between $\mathcal{B}^{\text{src}}$ and $\mathcal{B}^{\text{tar}}$. To address these challenges, VeoRL employs a three-stage approach (as detailed in Algorithm 1):

1. We introduce a *behavior abstraction network* designed to extract underlying control behaviors directly from the raw observations of unlabeled videos.

2. We design a hierarchical world model to capture state transitions in two ways: through real actions using a *trunk net* and through latent behaviors using a *plan net*.

3. We propose a model-based RL method that optimizes the target policy over the trunk net transitions while being guided by latent behavior rollouts from the plan net.

### 3.1. Latent Behavior Abstraction

To address the challenge of video data lacking action labels, previous methods (Baker et al., 2022; Zhang et al., 2022) typically focus on using target domain observations with real action labels (*i.e.,* the offline RL dataset) to train an inverse dynamics model, which is then used to assign pseudo-action labels to the source domain auxiliary videos. However, this approach assumes that the source and target domains share the same action space, leading to two potential issues: First, action predictions for source video data may suffer from domain distribution discrepancies. Second, the learned actions are limited in their ability to provide high-level behavior guidance for subsequent policy learning. Instead, we propose a fully unsupervised approach for behavior inference, leveraging a compact categorical latent action space significantly smaller than the raw continuous action space. This approach avoids the challenge of real action prediction, generating higher-level behavior representations that are distinct from the original actions for low-level visual control.

To obtain a discretized latent action space, we design a *behavior abstraction network* (BAN) based on vector quantization (Gray, 1984). As shown in Figure 2a, this module is trained within a one-step video prediction framework, where its output is used to drive a *recurrent state-space model* (RSSM) (Hafner et al., 2019) for forward dynamics modeling. The learning process begins by taking two observations $(o_{t-1}^{\text{tar}}, o_t^{\text{tar}})$ from the target dataset. We extract image encodings from these observations using a CNN model ($\text{E}_\theta$) and concatenate them into a latent vector, $[e_{t-1}^{\text{tar}}, e_t^{\text{tar}}] \in \mathbb{R}^D$, which captures the temporal difference between consecutive video frames. We maintain a learnable latent codebook $\mathbb{R}^{K \times D}$ consisting of $K$ continuous vectors $c_{k=1,...,K} \in \mathbb{R}^D$, each representing a distinct behavior abstraction. Given $[e_{t-1}^{\text{tar}}, e_t^{\text{tar}}]$, BAN identifies the nearest neighbor $c_i$ from the set $\{c_k\}_{k=1}^K$ and outputs this vector as the chosen latent

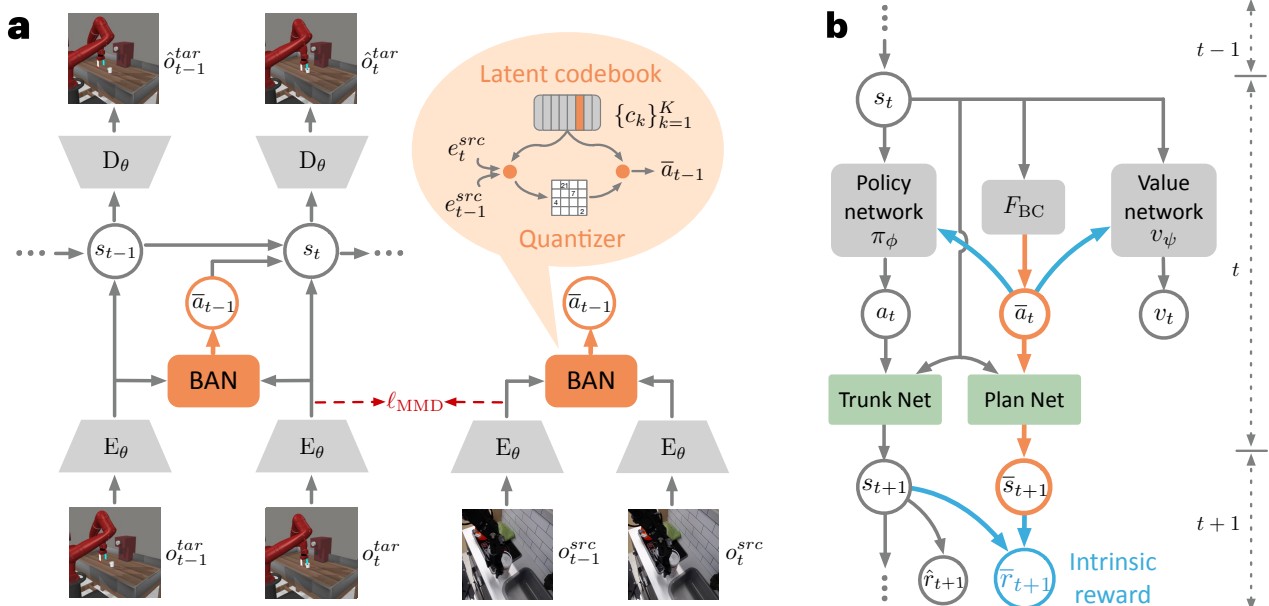

Figure 2. **Model architecture. a,** We construct a discrete, high-level latent action space by training the BAN, enabling forward dynamics modeling independent of real actions. **b,** The visualization of model-based actor-critic learning at a single rollout step. We leverage $F_{BC}$ to replay the video-informed latent behaviors, serving as the inputs of the actor and critic for producing goal-conditioned policies and value estimations, as well as the plan net for generating a long-term state rollout.

behavior $\bar{a}_{t-1}$, corresponding to the transition from $o_{t-1}^{tar}$ to $o_t^{tar}$. Similar to VQ-VAE (Van Den Oord et al., 2017), the optimization objective of the vector quantization process can be formulated as

$$\ell_{VQ} = \underbrace{\left\| \text{sg}[e_{t-1}^{tar}, e_t^{tar}] - c_i \right\|_2}_{\text{codebook loss}} + \underbrace{\left\| [e_{t-1}^{tar}, e_t^{tar}] - \text{sg}(c_i) \right\|_2}_{\text{commitment loss}}, \quad (1)$$

where $\text{sg}(\cdot)$ denotes gradient stopping. The codebook loss encourages the selected code to be closer to the input, while the commitment loss facilitates gradient backpropagation into the encoder. This helps stabilize the training process by preventing the image embedding from switching indiscriminately between different codes in the codebook.

As illustrated in Figure 2a, the selected latent behavior serves as input to a state transition model referred to as the *plan net*. We backpropagate the video prediction error to the BAN to refine the selection of latent behaviors, ensuring that they better predict subsequent frames. The overall objective function of BAN can be formulated as $\ell_{BAN} = \ell_{VQ} + \alpha \ell_{plan}$, where $\ell_{plan}$ denotes the loss function associated with the plan net, which will be detailed subsequently.

To leverage the BAN trained on $\mathcal{B}^{tar}$ as an estimator of latent behaviors for OOD videos from $\mathcal{B}^{src}$, we adopt the *maximum mean discrepancy* (MMD) loss (Borgwardt et al., 2006), a method widely used in domain adaptation (Huang et al., 2006; Li et al., 2013; Gong et al., 2013). The MMD loss is defined as $\ell_{MMD} = \left\| \mathbb{E}_{o_t^{src}} e_t^{src} - \mathbb{E}_{o_t^{tar}} e_t^{tar} \right\|_2^2$ and is applied to

the outputs of the image encoder to align visual embeddings across different domains.

### 3.2. Two-Stream World Model

We design a two-stream interactive world model consisting of: (i) a *Trunk net* that captures future state transitions driven by real actions and the associated environmental rewards, $(s_1, a_1, r_1, s_2, a_2, r_2, \ldots) \sim \text{Trunk}(o_1; \pi_\phi)$, and (ii) a *Plan net* that learns to predict future trajectories based on high-level behavior abstractions, $(\bar{s}_1, \bar{a}_1, \bar{s}_2, \bar{a}_2, \ldots) \sim \text{Plan}(o_1; \text{BAN})$. Here, $s_t = [h_t, z_t]$ and $\bar{s}_t = [\bar{h}_t, \bar{z}_t]$, where $(h_t, \bar{h}_t)$ and $(z_t, \bar{z}_t)$ represent the deterministic and stochastic components of the predicted states, respectively. Specifically, the plan net includes a behavior cloning module $F_{BC}$ that predicts $\bar{a}_t$ based solely on $\bar{s}_t$. The learning target for $F_{BC}$ is the latent behavior inferred by BAN from the state transition pair $(\bar{s}_t, \bar{s}_{t+1})$. The world model can be formulated as

Trunk Net:
$$\begin{cases} h_t = \text{GRU}(s_{t-1}, a_{t-1}; \theta) \\ z_t \sim p(h_t; \theta) \\ z_t' \sim q(h_t, e_t; \theta) \\ \hat{o}_t \sim p(s_t; \theta) \\ \hat{r}_t \sim p(s_t; \theta) \end{cases}$$

Plan Net:
$$\begin{cases} \bar{h}_t = \text{GRU}(\bar{s}_{t-1}, \bar{a}_{t-1}; \bar{\theta}) \\ \bar{z}_t \sim p(\bar{h}_t; \bar{\theta}) \\ \bar{z}_t' \sim q(\bar{h}_t, e_t; \bar{\theta}) \\ \bar{o}_t \sim p(\bar{s}_t; \theta) \\ \bar{a}_t = F_{BC}(\bar{s}_t; \bar{\theta}) \end{cases}$$

$$(2)$$

where $(z_t', \bar{z}_t')$ indicate the posteriors inferred from the current image observation and $(\hat{o}_t, \bar{o}_t)$ are the reconstructed

images. $\theta$ represents the combined parameters of the trunk net, while $\bar{\theta}$ represents the combined parameters of the plan net. The trunk net and plan net share the same model parameters for the image encoder and decoder, but have separate parameters for the remaining parts of the model. The trunk net is trained exclusively on the offline RL dataset $\mathcal{B}^{\text{tar}}$. The objective function is as follows:

$$\ell_{\text{trunk}}(\theta) = \mathbb{E}_{\tau \in \mathcal{B}^{\text{tar}}} \sum_{t=1}^{T} \underbrace{-\ln p(\hat{o}_t \mid s_t)}_{\text{image reconstruction}} \underbrace{-\ln p(\hat{r}_t \mid s_t)}_{\text{reward prediction}} \\ + \underbrace{\text{KL}[q(z_t' \mid h_t, e_t) \parallel p(z_t \mid h_t)]}_{\text{KL divergence}}. \quad (3)$$

The plan net is trained using trajectories from the target dataset $\mathcal{B}^{\text{tar}}$ and unlabeled videos from the source dataset $\mathcal{B}^{\text{src}}$. We adopt a specialized training strategy to ensure that the plan net models meaningful long-term state transitions. Instead of focusing on one-step future predictions, we consider the state transition "*shortcuts*" that occur when BAN predicts a different $\bar{a}_t$ compared to the previous step $\bar{a}_{t-1}$. Specifically, starting with an original trajectory $(o_1, \bar{a}_1, o_2, \bar{a}_2, \ldots, o_{T-1}, \bar{a}_{T-1}, o_T)$ sampled from the dataset and precomputed by BAN, we retain only those state-action pairs $(\bar{a}_t, o_{t+1})$ where a change in latent behavior is detected, i.e., , $\bar{a}_t \neq \bar{a}_{t-1}$. This process generates a refined trajectory that captures significant behavioral changes over time, represented as $(o_{j_1}, \bar{a}_{j_1}, o_{j_2}, \bar{a}_{j_2}, \ldots, o_{j_{n-1}}, \bar{a}_{j_{n-1}}, o_{j_n})$, where $o_{j_1} = o_1$. These refined trajectories are subsequently used to optimize the plan network through the following objective function:

$$\ell_{\text{plan}}(\bar{\theta}) = \mathbb{E}_{\tau \in \mathcal{B}^{\text{tar}}, \mathcal{B}^{\text{src}}} \sum_{t=j_1}^{j_n} \underbrace{-\ln p(\bar{o}_t \mid \bar{s}_t)}_{\text{image reconstruction}} \underbrace{-\ln p(\bar{a}_t \mid s_t)}_{\text{behavior cloning}} \\ + \underbrace{\text{KL}[q(\bar{z}_t' \mid \bar{h}_t, e_t) \parallel p(\bar{z}_t \mid \bar{h}_t)]}_{\text{KL divergence}}. \quad (4)$$

This approach enables the plan net to capture significant behavior shifts, enhancing its ability to predict impactful long-term transitions. It provides supplementary information that complements short-term, real action-based state transitions during the subsequent policy optimization stage.

### 3.3. Model-Based Policy Learning

We perform temporal difference learning using a probabilistic policy network $\pi_\phi(\cdot)$ and a deterministic value network $v_\psi(\cdot)$. As shown in Figure 2b, these model components are optimized over two-stream imagined trajectories generated by the trunk net and the plan net, i.e., $(s_1, \bar{s}_1, \hat{a}_1, \bar{a}_1, \hat{r}_1, s_2, \bar{s}_2, \hat{a}_2, \bar{a}_2, \hat{r}_2, \ldots, s_L, \bar{s}_L, \hat{a}_L, \bar{a}_L, \hat{r}_L)$, where $s_1$ is derived from an observation $o_1$ randomly sampled from the target set $\mathcal{B}^{\text{tar}}$.

Unlike previous methods, both the policy network and

the value network are additionally conditioned on the estimated latent behavior to incorporate high-level control guidance, such that $\pi_\phi(a_t \mid s_t, \bar{a}_t)$ and $v_\psi(s_t, \bar{a}_t)$, where $\bar{a}_t = F_{\text{BC}}(s_t)$. This modification allows for more informed decision-making by integrating abstract behavioral patterns from the auxiliary videos into the learning process.

The learning approach in Eq. (4) enables the plan net to predict long-term states that can serve as decision-making *subgoals*. Accordingly, we propose a goal-conditioned intrinsic reward that quantifies the similarity between short-term state transitions induced by real actions and long-term state transitions driven by high-level latent behaviors:

$$\begin{aligned} \bar{r}_t &= -\|s_t, \bar{s}_t\|_2, \\ s_t &\sim \text{Trunk}(s_{t-1}, \pi_\phi(s_{t-1}, \bar{a}_{t-1})), \\ \bar{s}_t &\sim \text{Plan}(s_{t-1}, F_{\text{BC}}(s_{t-1})). \end{aligned} \quad (5)$$

The policy network is trained to output actions that lead to states that maximize the value network's output, while the value network seeks to accurately estimate the discounted future returns achieved by the policy network starting from each imagined state. Specifically, the value target $V_t^\lambda$ for the value network incorporates a weighted average of reward information over an $n$-step future horizon, which is defined recursively as follows:

$$V_t^\lambda \doteq (\hat{r}_t + \omega \bar{r}_t) \\ + \gamma \begin{cases} (1 - \lambda)\, v_\psi(s_{t+1}, \bar{a}_{t+1}) + \lambda V_{t+1}^\lambda, & \text{if } t < L, \\ v_\psi(s_L), & \text{if } t = L. \end{cases} \quad (6)$$

where $\lambda$ is set to 0.95 for considering more on long horizon targets. $\gamma$ is a discount factor set to 0.99 in practice. Given a trajectory of model states and rewards, the value network $v_\psi$ is trained to regress $V_t^\lambda$ and the policy network $\pi_\phi$ is optimized with entropy regularization:

$$\ell(\psi) = \mathbb{E}_{p_\theta, p_\phi} \sum_{t=1}^{L-1} \frac{1}{2}(v_\psi(s_t, \bar{a}_t) - \text{sg}(V_t^\lambda))^2,$$

$$\ell(\phi) = \mathbb{E}_{p_\theta, p_\phi} \sum_{t=1}^{L-1} \Big( \underbrace{-(1-\rho)V_t^\lambda}_{\text{Dynamics BP}} \underbrace{-\eta H[a_t \mid s_t, \bar{a}_t]}_{\text{Entropy}} \quad (7) \\ \underbrace{-\rho \ln p_\phi(\hat{a}_t \mid s_t, \bar{a}_t)\text{sg}(V_t^\lambda - v_\psi(s_t, \bar{a}_t))}_{\text{REINFORCE}} \Big).$$

For the first term in the actor loss, we employ the straight-through gradients (Bengio et al., 2013) to backpropagate the value targets of the sampled actions and state sequences directly through the learned dynamics. In continuous control tasks like Meta-World and CARLA, we set $\rho = 0$. In discrete control tasks, we set $\rho = 1$ and observe that integrating REINFORCE gradients yields improved results.

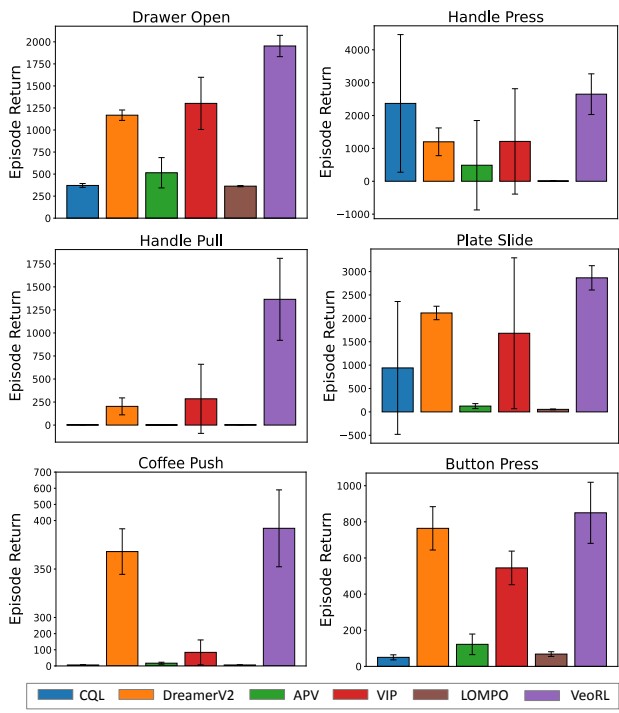

Figure 3. **Performance on Meta-World robotic manipulation tasks in episode return.** Error bars indicate standard deviation across the 50 evaluation episodes, with 3 random training seeds.

## 4. Experiments

In this section, we present (i) quantitative comparisons with existing RL methods on a diverse set of visual control benchmarks, (ii) offline-to-online transfer learning results on novel interactive control tasks, (iii) ablation studies for each proposed model component in VeoRL, and (iv) hyperparameter analyses along with visualizations of the learned latent behavior abstractions. The hyperparameters for implementing VeoRL are provided in Appendix Table 2.

To demonstrate the capability of VeoRL, we conduct experiments on three visual RL environments, including *Meta-World* (Yu et al., 2019), *CARLA* (Dosovitskiy et al., 2017), and *MineDojo* (Fan et al., 2022). Like D4RL (Fu et al., 2020), we collect the offline RL datasets of *medium*-quality trajectories using a partially-trained DreamerV2 agent (Hafner et al., 2020). Please refer to Appendix Section C for more details of the benchmarks.

We compare VeoRL with: (1) *DrQ+CQL* (Kumar et al., 2020) and *LOMPO* (Rafailov et al., 2021) are specifically designed for offline RL tasks. (2) *DreamerV2* (Hafner et al., 2020) serves as our primary baseline. It follows a similar model-based RL framework but lacks the video-enhancement method. (3) Like our approach, *APV* (Seo et al., 2022) and *VIP* (Ma et al., 2023) exploit auxiliary videos to improve model performance. (4) Finally,

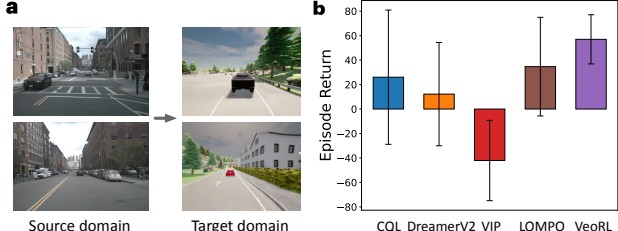

Figure 4. **Experiments of autonomous driving. a**, Showcases of the source NuScenes and target CARLA datasets. **b**, Performance comparison on CARLA, measured by averaged episode returns.

*VPT* (Baker et al., 2022) is particularly designed for Minecraft tasks, so we only use it as the baseline model on the MineDojo environment. Further details of the compared models can be found in Appendix Section D.

### 4.1. Main Results

**Meta-World robotic manipulation.** For the Meta-World offline RL tasks, we use the BridgeData-V2 dataset (Walke et al., 2023) with real-world videos as the unlabeled source domain. We conduct all tasks with 3 random seeds and report the mean results and standard deviations over 50 episodes. As shown in Figure 3, our approach achieves state-of-the-art performance in episodic returns over all six tasks in this domain. Specifically, VeoRL outperforms DreamerV2 by a large margin in *Drawer Open* (1168 → 1953), *Handle Press* (1201 → 2650) and *Handle Pull* (203 → 1365). Compared to APV and VIP, which also employ video-enhanced pretraining, VeoRL demonstrates a significant advantage by effectively exploiting and transferring the underlying control behaviors behind these videos. Furthermore, we provide additional model comparisons measured by success rate in Appendix Table 3, where VeoRL consistently achieves the best performance.

**CARLA autonomous driving.** We employ the large-scale NuScenes dataset (Caesar et al., 2019) with 1,000 diverse real-world driving scenes to improve the autonomous driving performance on the CARLA benchmark. Figure 4 shows the performance of our model compared to different baselines across three random training seeds. As we can see, our VeoRL achieves notably the best mean results with the smallest standard deviations, outperforming DreamerV2 by as much as **350%**. Notably, although the VIP method also involves pretraining on the same NuScenes dataset, its policy transfer approach (based on a self-supervised value-function objective) struggles to achieve satisfactory results in this driving task. We exclude the results from APV in Figure 4 to provide a clearer comparison of the other approaches, as it yields an average episode return of approximately −500.

**MineDojo open-world games.** MineDojo is an open-world 3D environment with a vast state space for the agent

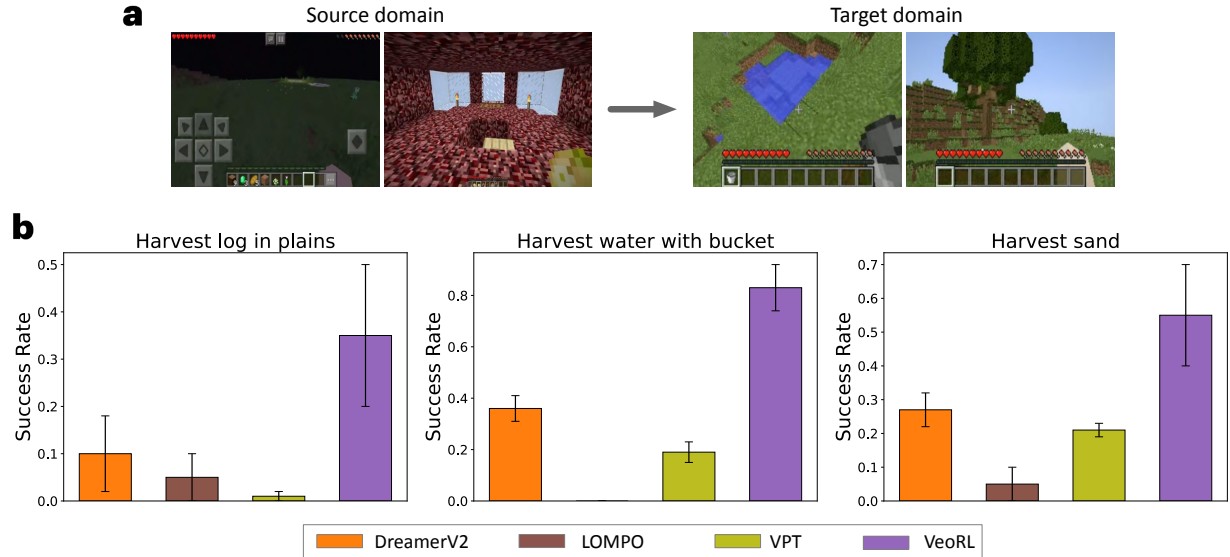

Figure 5. **Experiments of the MineDojo 3D navigation and control tasks. a**, Showcases of source online videos and target offline datasets. **b**, Performance comparison in success rate.

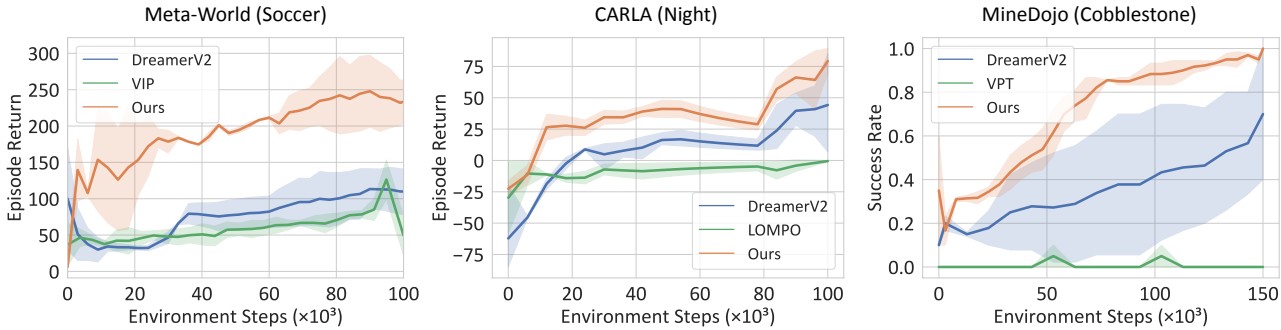

Figure 6. **Performance of offline-to-online finetuning on novel tasks.** By conducting offline pretraining, VeoRL demonstrates improved training efficiency and final performance when adapting to unseen, more challenging visual control tasks. Notably, we train the DreamerV2 model under the same offline-to-online setup. The results are averaged over 3 random training seeds.

to explore, making effective video demonstrations particularly essential for the target offline learning scenarios. As shown in Figure 5a, we use online videos of human players as the auxiliary source video set. These videos contain random Minecraft tasks that may differ significantly in gaming tactics from those for the target tasks. The results in Figure 5b show that VeoRL outperforms the other baseline models. Specifically, it improves upon DreamerV2 by **250%** ($0.10 \rightarrow 0.35$) in *Harvest log in plains*, by approximately **130%** ($0.36 \rightarrow 0.83$) in *Harvest water with bucket*, and by around **100%** ($0.27 \rightarrow 0.55$) in *Harvest sand*.

**Offline-to-online transfer learning.** To evaluate the generalizability of our model, we finetune the pretrained offline RL agents in *unseen* visual control tasks through online interactions with the environment. We select a more challenging *Soccer* task as our offline-to-online transfer target on the Meta-World benchmark. This task typically involves

a two-stage decision-making process where the agent must first fetch a ball and then push it to a goal position. For CARLA, the offline RL agents are pretrained using the *Day-mode* dataset and then finetuned in the *Night mode* driving scenarios. For MineDojo, the agents are pretrained on a fixed dataset for the *Harvest sand* task and finetuned on the *Harvest cobblestone with wooden pickaxe* task. As shown in Figure 6, VeoRL demonstrates superior training efficiency compared to DreamerV2, which follows the same offline-to-online domain adaptation setup, highlighting the effectiveness of our offline training approach.

### 4.2. Ablation Studies

As we will discuss in Section 3, VeoRL introduces three key innovations to the model-based actor-critic learning framework of DreamerV2 ([Hafner et al., 2021](#)): (i) It uses latent behaviors as inputs to the policy network (*i.e.,* actor); (ii) It

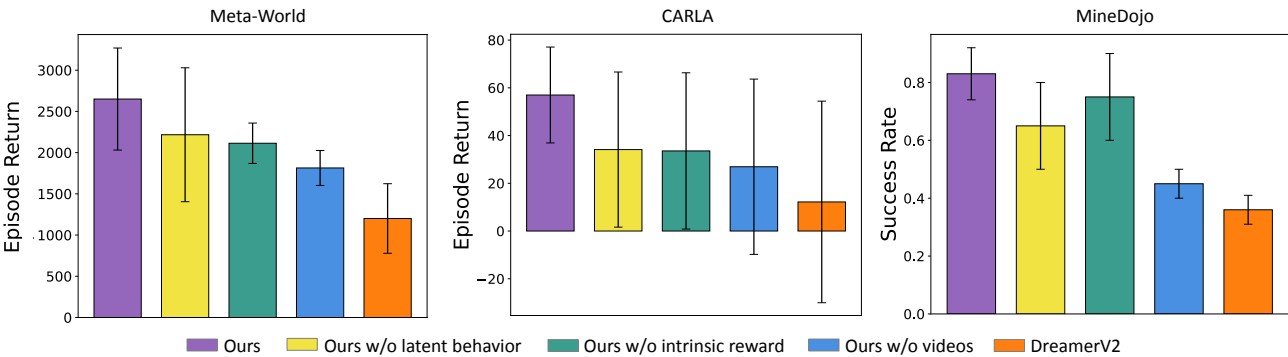

Figure 7. **Ablation studies of VeoRL.** We show the impact of latent behavior guidance (*yellow*) and the intrinsic reward (*green*). We also compare VeoRL with a baseline model that does not use any source video data throughout the training process (*blue*).

Table 1. **Performance comparison with DreamerV3 on Meta-World robotic manipulation tasks in success rate and episode return.** We compute the mean and standard deviation of 50 episodes over 3 random training seeds.

| Methods | DreamerV2 | DreamerV3 | VeoRL(DV2) | VeoRL(DV3) |
|---|---|---|---|---|
| | | Success Rate | | |
| Drawer Open | $0.18 \pm 0.04$ | $0.00 \pm 0.00$ | $0.70 \pm 0.07$ | $0.55 \pm 0.15$ |
| Handle Press | $0.33 \pm 0.11$ | $0.05 \pm 0.05$ | $0.60 \pm 0.12$ | $0.35 \pm 0.15$ |
| | | Episode Return | | |
| Drawer Open | $1168.35 \pm 59.55$ | $674.55 \pm 79.04$ | $1953.60 \pm 121.48$ | $1393.50 \pm 122.50$ |
| Handle Press | $1201.75 \pm 422.10$ | $257.85 \pm 247.05$ | $2650.90 \pm 619.60$ | $1360.15 \pm 547.85$ |

employs latent behaviors to compute an intrinsic reward for training the value network (*i.e.,* critic), and (iii) It trains the *plan net* in the world model using the auxiliary video set and the offline RL dataset jointly. Accordingly, we conduct ablation studies to evaluate the effectiveness of individual techniques. Figure 7 provides corresponding results on Meta-World (*Handle Press*), CARLA and MineDojo (*Harvest water with bucket*). As illustrated by the yellow and green bars, excluding the latent behavior input in the policy network or removing its influence from the value network's training objective both result in significant performance drops in VeoRL. These results highlight the crucial role of latent behaviors as effective high-level guidance in policy learning. The blue bar illustrates that training all network components solely on the target offline dataset (without access to auxiliary videos) leads to a significant performance decline. This highlights that our video-enhanced method effectively extracts valuable prior knowledge of underlying expert behaviors from the unlabeled videos.

### 4.3. Comparison with DreamerV3

The decision to use DreamerV2 as the backbone is driven by two key considerations: (i) *Offline setting performance*. While DreamerV3 (Hafner et al., 2023) achieves strong performance across diverse tasks with fixed hyperparameters,

our experiment results in Table 1 demonstrate that it underperforms in offline RL settings without hyperparameter tuning compared to DreamerV2. (ii) *Backbone consistency*. LAMPO, a key baseline in our work, uses DreamerV2 as its backbone. To ensure direct and fair comparisons between our method and LAMPO, we maintain consistency by adopting the same backbone. Notably, we also conduct additional experiments on Meta-World by integrating VeoRL with DreamerV3. The results demonstrate that our approach outperforms vanilla DreamerV3, showcasing its ability to seamlessly integrate with different architectures.

### 4.4. Analyses of Latent Actions

To illustrate the correspondence between latent behaviors and real actions, we evaluate the distribution of the learned latent behavior abstractions in Meta-World *Coffee Push* in Figure 8. In VeoRL, we roll out the trunk net and plan net simultaneously for model-based policy learning. Starting from the same input state $s_t$, we obtain two future trajectories of real actions $a_{t:t+H}$ and corresponding latent behaviors $\bar{a}_{t:t+H}$. To show the correspondence between them, we specifically select trajectories of four consecutive real actions $a_{t:t+4}$ with time-invariant latent behaviors $\bar{a}_t = \bar{a}_{t:t+4}$, and then use the t-SNE algorithm (Van der Maaten & Hinton, 2008) to project $a_{t:t+4}$ into a 2D plane.

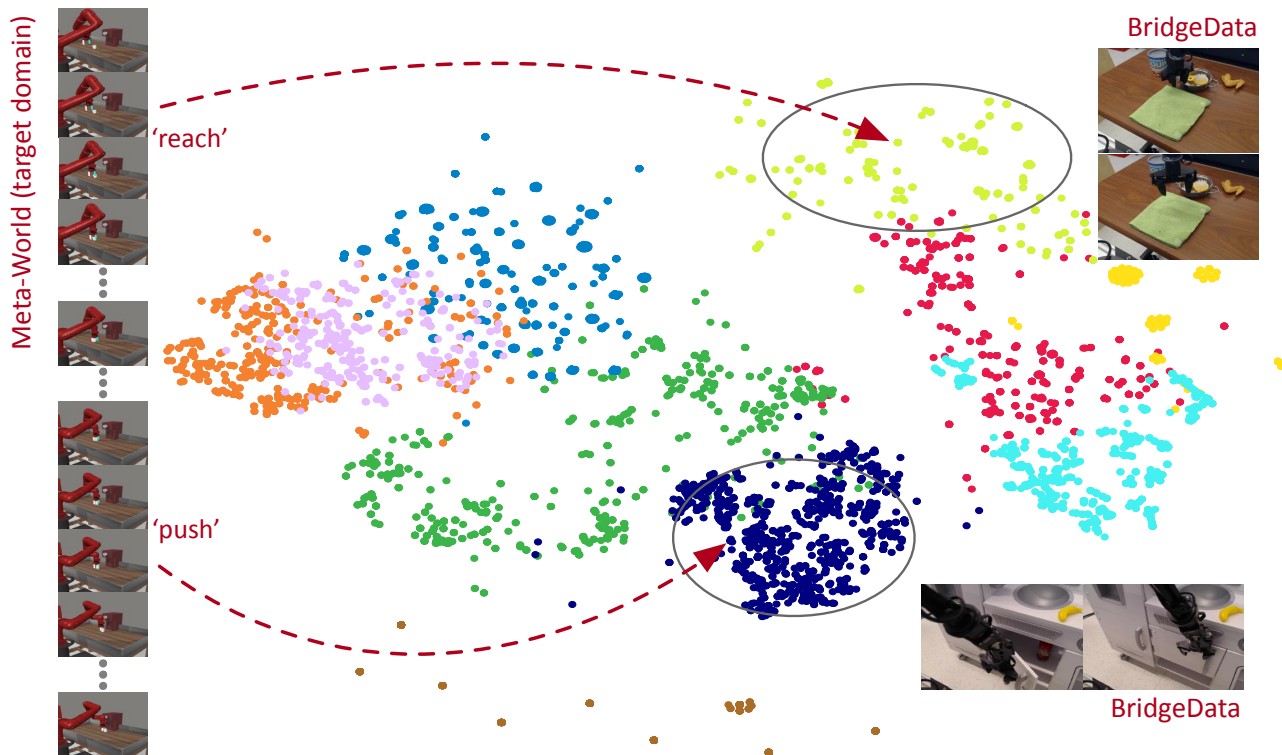

*Figure 8.* **Correspondence between latent behaviors and real actions in Meta-World.** During model-based state rollouts, starting from the same state $s_t$ of the target domain, VeoRL simultaneously generates real action trajectories $a_{t:t+H}$ and latent behaviors $\bar{a}_{t:t+H}$. We consider trajectories of four consecutive real actions $a_{t:t+4}$, where the latent behaviors are time-invariant, *i.e.,* $\bar{a}_t = \bar{a}_{t:t+4}$. We then use t-SNE to project $a_{t:t+4}$ into a 2D plane. To distinguish between different latent behaviors, we assign distinct colors to the points, visualizing 10 latent behaviors for clarity. We present BridgeData examples where VeoRL generates the corresponding latent behaviors.

Different colors assigned to the points indicate distinct latent behaviors. From Figure 8, we observe that the latent behaviors align well with distinct clusters of real action sequences, demonstrating the high representational fidelity of the learned latent action space. These latent behaviors effectively capture semantic information from online videos and can be interpreted as fundamental operational skills, which can then be used as policy guidance for the RL agent. Furthermore, we provide the ablation results of varying the number of discrete latent behavior in Appendix Figure 9.

### 4.5. Further Results and Discussion

Please refer to Appendices F-I for additional experimental results, including the quality analysis of source videos and target domain trajectories, the hyperparameter analysis of latent action space, the generalizability across tasks, and the impact of MMD in domain adaptation.

## 5. Conclusions and Limitations

In this paper, we introduced VeoRL, a novel approach that leverages freely available, unlabeled video data to enhance offline visual RL. VeoRL constructs a world model with two state transition branches: one conditioned on latent behaviors extracted from video data and the other on the agent's real actions. This dual-branch structure allows the agent to effectively learn control policies by aligning state rollouts from two branches, thereby transferring knowledge from the diverse, out-of-domain video data to the RL agents. Our results demonstrate that VeoRL significantly outperforms existing offline RL methods across a wide range of challenging visual control tasks, including robotic manipulation, autonomous driving, and open-world video games. The integration of video data not only boosts the agent's ability to generalize across different domains but also accelerates the learning process by incorporating a broader understanding of the physical world and control policies.

The limitation of this work is the computational overhead. The training process of VeoRL involves extracting latent behaviors and optimizing a world model with two state transition branches. Despite the performance gain, the dual-branch setup could increase the computational complexity and memory requirements, making the approach more resource-intensive than simpler RL algorithms.

## Acknowledgments

This work was supported by the National Natural Science Foundation of China (62250062), the Smart Grid National Science and Technology Major Project (2024ZD0801200), the Shanghai Municipal Science and Technology Major Project (2021SHZDZX0102), and the Fundamental Research Funds for the Central Universities.

## Impact Statement

VeoRL represents a significant advancement in offline RL by leveraging unlabeled video data to enhance agent policy learning and generalization. It enables agents to be trained on vast amounts of real-world data without the need for labor-intensive annotations, making it more accessible for researchers, developers, and practitioners. VeoRL brings RL closer to how humans learn by observing and interacting with the world. Additionally, by aligning RL with human learning processes, VeoRL enables the development of more intuitive AI systems.

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

# Appendix

## A. Algorithm of VeoRL

We provide the overall training scheme of VeoRL in Algorithm 1.

---

**Algorithm 1** The training scheme of VeoRL

---

**Require:** Target offline RL dataset $\mathcal{B}^{\text{tar}}$ and source video set $\mathcal{B}^{\text{src}}$.
**Initialize:** Random policy network $\pi_\phi$ and value network $v_\psi$.

 1: **while** not converged **do**
 2:     `// Behavior abstraction learning`
 3:     **for** update $m_1$ steps **do**
 4:         Draw observation sequences $o_{1:T}^{\text{tar}} \sim \mathcal{B}^{\text{tar}}$ and $o_{1:T}^{\text{src}} \sim \mathcal{B}^{\text{src}}$.
 5:         Train BAN using $\ell_{\text{VQ}} + \alpha \ell_{\text{plan}}$ on $o_{1:T}^{\text{tar}}$.
 6:         Train the image encoder using $\ell_{\text{MMD}}$ on $o_{1:T}^{\text{tar}}$ and $o_{1:T}^{\text{src}}$.
 7:     **end for**
 8:     `// World model learning`
 9:     **for** update $m_2$ steps **do**
10:         Draw trajectories $\tau^{\text{tar}} = \{(o_t^{\text{tar}}, a_t^{\text{tar}}, r_t^{\text{tar}})\}_{t=1}^T \sim \mathcal{B}^{\text{tar}}$ and $o_{1:T}^{\text{src}} \sim \mathcal{B}^{\text{src}}$.
11:         Train the low-level trunk net using Objective (3) on $\tau^{\text{tar}}$.
12:         Predict latent behaviors $\bar{a}_{1:T-1}^{\text{tar}} \sim \text{BAN}(o_{1:T}^{\text{tar}})$ and $\bar{a}_{1:T-1}^{\text{src}} \sim \text{BAN}(o_{1:T}^{\text{src}})$.
13:         Train the high-level plan net using Objective (4) on $(o_{1:T}^{\text{tar}}, \bar{a}_{1:T-1}^{\text{tar}})$ and $(o_{1:T}^{\text{src}}, \bar{a}_{1:T-1}^{\text{src}})$.
14:         Train the behavior cloning network $F_{\text{BC}}$ using $\bar{a}_{1:T-1}^{\text{tar}}$ and $\bar{a}_{1:T-1}^{\text{src}}$.
15:     **end for**
16: **end while**
17: `// Model-based policy learning`
18: **while** not converged **do**
19:     Draw $\{(o_t^{\text{tar}}, a_t^{\text{tar}}, r_t^{\text{tar}})\}_{t=1}^T \sim \mathcal{B}^{\text{tar}}$.
20:     **for** each $t$ **do**
21:         Generate $\{(s_t, a_t, r_t)\}_t^{t+L}$ using $\pi_\phi$ and the low-level trunk net.
22:         Generate $\{(\bar{s}_t, \bar{a}_t)\}_t^{t+L}$ using $F_{\text{BC}}$ and the high-level plan net.
23:         Compute the intrinsic rewards $\{\bar{r}_t\}_t^{t+L}$ using Eq. (5).
24:         Update $v_\psi$ and $\pi_\phi$ using Objective (7).
25:     **end for**
26: **end while**

---

## B. Implementation Details

The model configurations and hyperparameter settings are detailed in Table 2.

## C. Benchmark Details

We evaluate the performance of VeoRL in the following three visual RL environments:

- **Meta-World** (Yu et al., 2019): The Meta-World benchmark simulates 50 manipulation tasks with complex visual dynamics, all executed by the same robotic arm. These tasks involve a variety of actions such as reaching, pushing, and grasping, implemented with the MuJoCo physics engine (Todorov et al., 2012). The action space is represented as a 2-tuple: the 3D movement of the end-effector and a normalized torque for the gripper fingers, with values ranging from $-1$ to 1. We collect six offline datasets from the tasks *button press*, *drawer open*, *handle pull*, *handle press*, *plate slide*, and *coffee push*. Each dataset consists of 200 trajectories, each with 500 time steps.

- **CARLA** (Dosovitskiy et al., 2017): CARLA provides realistic visual observations for autonomous driving research, including varied weather conditions, road layouts, and traffic signs. The agent controls the vehicle using steering, acceleration, and braking commands. The goal is to maximize driving distance over 1,000 time steps while avoiding

*Table 2.* An overview of layers and hyperparameters used in VeoRL in the three environments.

| Item | Meta-World | CARLA | MineDojo |
|---|---|---|---|
| World Model | | | |
| Image encoder | Conv3-32 | Conv3-32 | Conv3-96 |
| GRU hidden size | 200 | 200 | 4096 |
| RSSM number of units | 200 | 200 | 1024 |
| Stochastic latent dimension | 50 | 50 | 32 |
| Discrete latent classes | 0 | 0 | 32 |
| Weighting factor $\alpha$ | 1 | 1 | 1 |
| World model learning rate | $3 \cdot 10^{-4}$ | $3 \cdot 10^{-4}$ | $1 \cdot 10^{-4}$ |
| BAN's update iterations | 40K | 40K | 30K |
| Behavior Learning | | | |
| Imagination horizon $L$ | 15 | 15 | 15 |
| $\lambda$-target | 0.95 | 0.95 | 0.95 |
| Discount $\gamma$ in Eq. (6) | 0.99 | 0.99 | 0.99 |
| $\omega$ in Eq. (6) | 0.05 | 0.1 | $1 \cdot 10^{-5}$ |
| $\rho$ in Eq. (7) | 0 | 0 | 1 |
| $\eta$ in Eq. (7) | $1 \cdot 10^{-4}$ | $1 \cdot 10^{-4}$ | $3 \cdot 10^{-4}$ |
| MLP number of Policy network | 4 | 4 | 5 |
| MLP number of Value network | 3 | 3 | 5 |
| Policy network learning rate | $8 \cdot 10^{-5}$ | $8 \cdot 10^{-5}$ | $3 \cdot 10^{-5}$ |
| Value network learning rate | $8 \cdot 10^{-5}$ | $8 \cdot 10^{-5}$ | $3 \cdot 10^{-5}$ |
| Environment Setting | | | |
| Time limit | 500 | 1000 | 1000 |
| Action repeat | 1 | 4 | 1 |
| Image size | $64 \times 64$ | $64 \times 64$ | $64 \times 64$ |

collisions with 30 vehicles or barriers. The reward function incorporates velocity, collision impact, and steering effort, encouraging safe and efficient driving. The offline dataset includes approximately 1,000 episodes.

- **MineDojo** (Fan et al., 2022): This platform provides convenient APIs built on Minecraft, standardizing task specifications, world settings, and observation and action spaces for agents. In our study, we focus on three challenging tasks: *harvest log in plains*, *harvest water with bucket*, and *harvest sand*. We set the maximum time steps of each episode as 1,000 and adopt a binary reward indicating task completion, supplemented by the MineCLIP reward (Fan et al., 2022).

For the auxiliary video datasets, we select two datasets, BridgeData-V2 and NuScenes, as source domains due to their relevance to Meta-World and CARLA, as well as publicly available Minecraft videos for MineDojo:

- **BridgeData-V2** (Walke et al., 2023): This is a large and diverse robot manipulation dataset featuring 24 environments and 60,096 trajectories. Most trajectories are video recordings from the Toy Kitchen and Toy Tabletops environments. For our experiments, we use trajectories longer than 20 steps as the source domain data for Meta-World, resulting in approximately 50,000 trajectories.

- **NuScenes** (Caesar et al., 2019): This large-scale autonomous driving dataset contains 1,000 diverse scenes with detailed 3D object annotations. For our experiments, we use only the video recordings, excluding annotations, as the source domain data for tasks in the CARLA environment. In total, we collect 850 episodes with varying time step lengths.

- **Minecraft videos:** For the source domain of MineDojo, we use publicly available videos of humans playing Minecraft. We collect a dataset of 5,000 episodes with varying lengths.

*Table 3.* **Performance on Meta-World robotic manipulation tasks in success rate.** We compute the mean and standard deviation of 50 episodes over 3 random training seeds.

| Methods | DrQ + CQL | DreamerV2 | APV | VIP | LOMPO | Ours |
|---|---|---|---|---|---|---|
| Drawer Open | $0 \pm 0$ | $0.18 \pm 0.04$ | $0.02 \pm 0.05$ | $0.02 \pm 0.04$ | $0 \pm 0$ | $\mathbf{0.70 \pm 0.07}$ |
| Handle Press | $0.60 \pm 0.48$ | $0.33 \pm 0.11$ | $0.10 \pm 0.09$ | $0.56 \pm 0.17$ | $0 \pm 0$ | $\mathbf{0.60 \pm 0.12}$ |
| Handle Pull | $0 \pm 0$ | $0.04 \pm 0.01$ | $0 \pm 0$ | $\mathbf{0.36 \pm 0.15}$ | $0 \pm 0$ | $0.30 \pm 0.10$ |
| Plate Slide | $0.23 \pm 0.41$ | $0.48 \pm 0.02$ | $0 \pm 0$ | $0.34 \pm 0.12$ | $0 \pm 0$ | $\mathbf{0.67 \pm 0.03}$ |
| Coffee Push | $0 \pm 0$ | $0.29 \pm 0.01$ | $0.12 \pm 0.11$ | $0.16 \pm 0.15$ | $0 \pm 0$ | $\mathbf{0.31 \pm 0.05}$ |
| Button Press | $0 \pm 0$ | $0.58 \pm 0.08$ | $0.01 \pm 0.01$ | $0.18 \pm 0.08$ | $0 \pm 0$ | $\mathbf{0.62 \pm 0.02}$ |
| Average | $0.14$ | $0.32$ | $0.04$ | $0.27$ | $0$ | $\mathbf{0.53}$ |

*Table 4.* **The ablation study of the quality of source videos on Meta-World (Handle Press).** As the number of source domain videos increases, the model's performance improves accordingly.

| Handle Press | All videos | 1/2 videos | 1/4 videos | DreamerV2 |
|---|---|---|---|---|
| Episode return | $2651 \pm 620$ | $2477 \pm 441$ | $1859 \pm 423$ | $1202 \pm 422$ |

## D. Compared Methods

We compare VeoRL against existing model-based and model-free offline RL methods:

- **CQL** (Kumar et al., 2020): A model-free method that conservatively learns value functions to address value overestimation. Following (Fu et al., 2020), we integrate CQL regularizers into the DrQ-V2 algorithm (Yarats et al., 2021a) to adapt it for pixel-based inputs.

- **DreamerV2** (Hafner et al., 2019): A model-based RL approach that learns policies directly from latent states within the world model. We adapt it to the offline setting and refer to this version as Offline DV2.

- **APV** (Seo et al., 2022): This method builds an action-conditional RSSM model on top of an action-free RSSM model pretrained on video datasets. We adapt APV to the offline scenario for a fair comparison.

- **VIP** (Ma et al., 2023): A pretraining approach that focuses on learning universal visual representations from video datasets, enabling generalization to downstream offline tasks. For this comparison, we pair VIP with the AWR algorithm (Peng et al., 2019) for offline policy learning.

- **LOMPO** (Rafailov et al., 2021): A model-based offline visual RL method that addresses model uncertainty in the latent space while incorporating explicit reward estimation.

- **VPT** (Baker et al., 2022): A foundational model for Minecraft trained using standard behavior cloning. VPT uses an inverse dynamics model trained on limited labeled data to annotate a vast set of unlabeled online videos. For task-specific performance, we finetune VPT on the offline dataset using PPO (Schulman et al., 2017), following the methodology outlined in DECKARD (Nottingham et al., 2023).

## E. Further Results of Performance Comparison

Table 3 provides a detailed comparison of VeoRL and existing methods on the Meta-World benchmark, evaluated by success rate. VeoRL outperforms DreamerV2 with a remarkable advantage in *Drawer Open* ($0.18 \rightarrow 0.70$), *Handle Pull* ($0.04 \rightarrow 0.30$) and *Handle Press* ($0.33 \rightarrow 0.60$) in terms of success rate.

## F. Analyses of the Quality of Source Videos and Target Domain Trajectories

To partially investigate the impact of using fewer high-quality videos, we have conducted an ablation study analyzing how the quantity of unlabeled source videos affects policy performance. The results are summarized in Table 4. As the number of

source domain videos increases, the model's performance improves accordingly, demonstrating that our method effectively leverages information from source domain videos and exhibits strong scalability. Even when using only a quarter of the videos, VeoRL's results still significantly outperform DreamerV2 (which does not utilize any source domain videos for training), indicating that our method can effectively extract useful skills from the videos.

Our framework trains the Plan Net and the corresponding BC model using data from both the source domain (BridgeData) and the target domain (Meta-World). This design leverages the diversity of BridgeData to enable the BC model to infer high-level behavioral abstractions (*e.g.*, "reach" or "grasp") that are applicable to the target task (*e.g.*, "Button press"), even when the offline data is of low quality and lacks successful trajectories. To validate this, we conduct a new experiment in which the offline Meta-World data consists of random trajectories, potentially containing extremely poor demonstrations. As shown in Table 5, the results indicate that our method could still benefit from the latent skills (estimated by BC) from the auxiliary videos, demonstrating the robustness of our approach in challenging offline RL settings.

*Table 5.* **The ablation study of the quality of target domain trajectories on Meta-World (Button Press) in episode return.** It demonstrates the robustness of our approach in challenging offline RL settings.

| Button Press | Medium offline data | Random offline data |
|---|---|---|
| Ours | $850 \pm 169$ | $638 \pm 150$ |
| DreamerV2 | $765 \pm 120$ | $505 \pm 150$ |
| VIP | $545 \pm 94$ | $276 \pm 99$ |

## G. Hyperparameter Analysis of Latent Action Space

Figure 9 provides the results of varying the number of discrete latent behaviors in the Meta-World (*Handle Press*) and CARLA benchmarks. We observe that setting $num = 50$ consistently yields the best performance across all tested environments, striking a good balance between the complexity of the learned latent action space and its stability of policy learning. Furthermore, VeoRL demonstrates robust performance across varying numbers of latent behaviors, compared with DreamerV2 without training with natural videos from the BridgeData-V2 and NuScenes datasets (dashed lines).

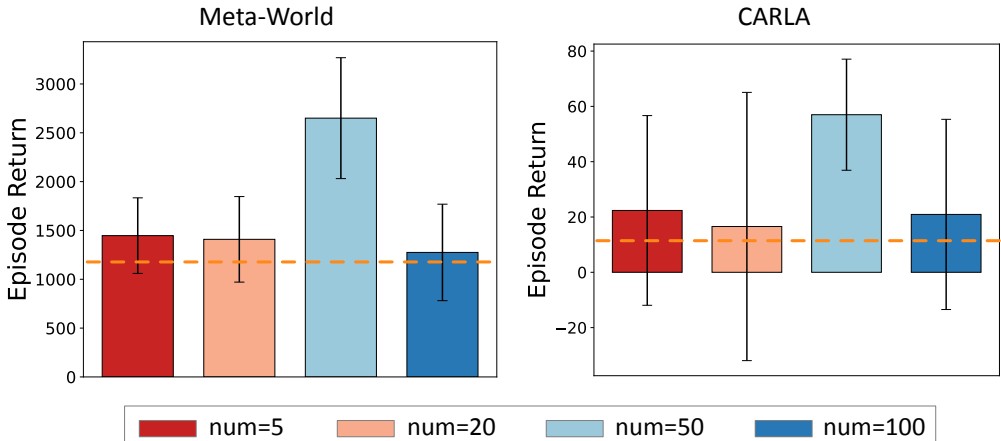

*Figure 9.* **Hyperparameter analyses of the dimension of the latent action space.** We observe that (i) VeoRL demonstrates robust performance across varying numbers of latent actions, compared with DreamerV2 without training with natural videos from the BridgeData-V2 and NuScenes datasets; (ii) A setting of 50 latent actions consistently yields the best performance across different tasks.

## H. Generalizability across Tasks

We conduct the experiments on Meta-World and MineDojo to validate the transferability of latent behaviors across tasks. In the training phase, the behavior abstraction network is frozen after training on Task A, with a fixed latent codebook. For Task B, this pre-trained network is directly deployed without undergoing task-specific fine-tuning on source video data. As presented in Table 6, the minimal performance degradation confirms that our method's latent behaviors trained on Task A

remain effective for Task B, consistently outperforming other baselines trained specifically on Task B. This transferability significantly reduces training costs for downstream tasks, as it spares the time for re-training on the auxiliary videos.

*Table 6.* **The study of the generalization ability of the latent behaviors across tasks on Meta-World and MineDojo.** We compute the mean and standard deviation over 3 random training seeds.

| Methods | Codebook construction | Downstream task | Success rate |
|---------|----------------------|-----------------|--------------|
| | Meta-World | | |
| VeoRL | Button Press | Button Press | $0.62 \pm 0.02$ |
| VeoRL | Handle Press | Button Press | $0.63 \pm 0.12$ |
| DreamerV2 | N/A | Button Press | $0.58 \pm 0.08$ |
| VIP | N/A | Button Press | $0.18 \pm 0.08$ |
| | MineDojo | | |
| VeoRL | Harvest sand | Harvest sand | $0.55 \pm 0.05$ |
| VeoRL | Harvest water with bucket | Harvest sand | $0.50 \pm 0.10$ |
| DreamerV2 | N/A | Harvest sand | $0.25 \pm 0.06$ |
| VPT | N/A | Harvest sand | $0.20 \pm 0.04$ |

# I. Analyses of MMD in Domain Adaptation

We performed an ablation study in the Meta-World environment to evaluate the role of MMD loss. To quantify its impact, we compared performance under conditions with and without MMD loss, accompanied by feature distribution visualizations derived from principal component analysis (PCA). The results in Table 7 validate that MMD loss critically bridges domain discrepancies. By aligning feature distributions, it enables the target domain to effectively transfer and adapt knowledge from the source domain, leading to a 33% improvement in success rates (0.60 vs. 0.45). Furthermore, as shown in the Figure 10, incorporating MMD significantly aligns the feature distributions between the source and target domains. This alignment correlates with policy generalization.

*Table 7.* **The ablation study on the MMD loss in the Meta-World environment.**

| Handle Press | w/ MMD loss | w/o MMD loss | DreamerV2 |
|--------------|-------------|--------------|-----------|
| Episode return | $2651 \pm 620$ | $1961 \pm 585$ | $1202 \pm 422$ |
| Success rate | $0.60 \pm 0.12$ | $0.45 \pm 0.15$ | $0.33 \pm 0.11$ |

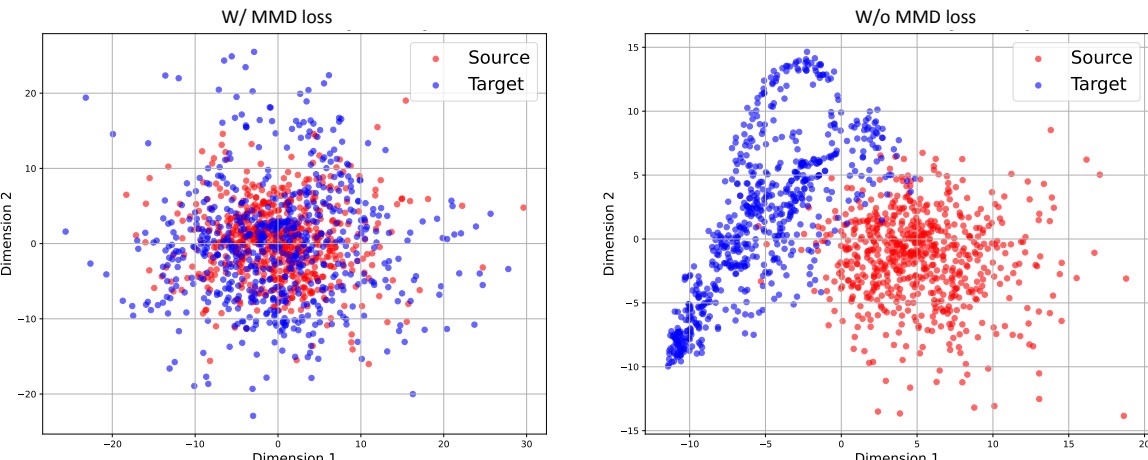

*Figure 10.* **Distribution visualization using PCA.** We observe that incorporating MMD significantly aligns the feature distributions between the source and target domains.

