# OpenReview forum: "Video-Enhanced Offline Reinforcement Learning: A Model-Based Approach"
_ICML.cc/2025/Conference — ICML 2025 poster_

### Official Review · Reviewer_dg3G · 2025-03-10

**Overall Recommendation:** 3

**Summary:**

The paper proposes a method that leverages unlabeled internet videos to enhance the performance of offline RL. A discretized latent action space is learned on unlabeled video data is learned by the behavior abstraction network (BAN) using vector quantization. The paper proposes using a two-stream world model, one conditioned on latent behaviors extracted from video data and the other on the agent’s real actions. This dual-branch structure allows the agent to effectively learn control policies by aligning state rollouts from two branches, thereby transferring knowledge from the diverse, out-of-domain video data to the RL agents. The authors compare with several prior works to show the efficacy of the proposed method. The paper also provides an ablation study to justify the design choices.

**Claims And Evidence:**

Yes.

**Essential References Not Discussed:**

Being a researcher in a closely related but not the same field, the references seem adequate to me.

**Experimental Designs Or Analyses:**

Overall, the experiments and analyses seem comprehensive.

**Methods And Evaluation Criteria:**

Yes

**Other Comments Or Suggestions:**

- It would be great if the authors could include an ablation study of the effect of different amounts of unlabeled data on policy performance. It would be interesting to see if reducing the amount of data also reduces the policy performance.

**Other Strengths And Weaknesses:**

Strengths
- The paper tackles an important problem of enhancing offline learning with unlabeled video data and shows the efficacy of the proposed approach in improving offline RL.
- The proposed method significantly improves upon prior work and also enables finetuning online on a new scenario with better sample efficiency than DreamerV2.
- The authors also provide an ablation study justifying design choices.

Weaknesses
- The authors use Dreamer V2 as their primary baseline. However, this is an outdated baseline and it would be great if the authors could provide comparisons with Dreamer V3.
- It would be great if the authors could include details about how Dreamer V2, which is an online RL algorithm, is adapted to the offline setting.

**Questions For Authors:**

It would be great if the authors could address the questions from the previous section and in Weaknesses.

**Relation To Broader Scientific Literature:**

The paper proposes a method for enhancing offline RL with off-domain video data. This is an important direction and baking in task priors from internet videos could potentially improve the robustness as well as sample efficiency of such policies.

**Theoretical Claims:**

No theoretical claims.

---

> ### Author Rebuttal · Authors · 2025-03-31
>
> We thank the reviewer for the insightful comments. Below, we address each comment point-by-point.
>
> > Q1. The authors use Dreamer V2 as their primary baseline. However, this is an outdated baseline and it would be great if the authors could provide comparisons with Dreamer V3.
>
> Please refer to our response to **Reviewer LSCr Q1**.
>
> > Q2. It would be great if the authors could include details about how Dreamer V2, which is an online RL algorithm, is adapted to the offline setting.
>
> The core adaptation of DreamerV2 to the offline setting lies in how the replay buffer is initialized and utilized. In the original online DreamerV2, the agent interacts with the environment to collect new trajectories incrementally. These trajectories are stored in a replay buffer, and the world model and policy are trained by sampling batches from this dynamically updated buffer. In the offline setting, we instead preload the entire static dataset into the buffer once at the start of training. No new data is collected during training, and all models are trained exclusively on this fixed dataset.
>
> > Q3. It would be great if the authors could include an ablation study of the effect of different amounts of unlabeled data on policy performance. It would be interesting to see if reducing the amount of data also reduces the policy performance.
>
> As suggested, we have conducted a new ablation study to analyze how the amount of unlabeled data impacts policy performance. The results are summarized below. As the amount of source domain data increases, the model's performance improves accordingly, demonstrating that our method effectively leverages information from source domain videos and exhibits strong scalability. Full results across different datasets will be included in the revised paper.
>
> |Handle Press   | All videos  | 1/2 videos | 1/4 videos | w/o videos (results from Fig 7) | DreamerV2 |
> | ----- | --- | --- |--- | --- |--- |
> |Episode return  |  2651 $\pm$ 620      | 2477 $\pm$ 441      |  1859 $\pm$ 423 | 1814 $\pm$ 212.06 |  1201.75 $\pm$ 422.10|

---

### Official Review · Reviewer_W3sg · 2025-03-10

**Overall Recommendation:** 3

**Summary:**

This work proposes a novel method called VeoRL (Video-enhances offline RL). This method allows to use additional pre-training video data, without the need for reward or action annotations. That data is used to train a behavior abstraction network and the planning network. The in-domain data is used to train the planning network, as well as the trunk network. Then, inside the world model, a policy is trained to optimize the rewards, as well as to optimize suboal reaching objective, with subgoals specified by the planning network. The method is evaluated on metaworld, Carla and minecraft, showing impressive performance compared to existing methods such as APV, VIP, and DreamerV2.

**Claims And Evidence:**

The main claim:  VeoRL can transfer common sense knowledge of control behaviors from natural videos.

The claim is supported by the experiments.

**Essential References Not Discussed:**

This paper doesn't cite FICC, which is another method that tackles a very similar problem:

Ye, Weirui, et al. "Become a proficient player with limited data through watching pure videos." _The Eleventh International Conference on Learning Representations_. 2022.

**Experimental Designs Or Analyses:**

I checked the design of the main expeirments. One issue I have is the choice of DreamerV2 as opposed to DreamerV3. DreamerV3 appears to be a more natural baseline in this case, especially for minecraft, as it has demonstrated good performance there. Can the authors justify their choice of DreamerV2 over V3? Is that due to the fact that the world models you trained have the DreamerV2-style architecture?

**Methods And Evaluation Criteria:**

Yes.

**Other Comments Or Suggestions:**

247 right - considering more on -> did you mean "focusing more on"?

**Other Strengths And Weaknesses:**

###### Strengths
- The paper investigates an important yet underexplored direction of research, an introduces a novel method
- The proposed method shows great improvement over the baselines on the proposed experiments
- The paper is well-written

###### Weaknesses
- The method is compared to DreamerV2 and not V3
- The tasks selected for minecraft and metaworld are quite simple, and do not involve multi-step decision making, e.g. pick and place for metaworld, or crafting an metal sword for minecraft
- The natural videos this method leverages contain, for the most part, high-quality trajectories. This enables the planner to simply use a behavior cloning policy. This however is a quite limiting assumption.

**Questions For Authors:**

Q1 Why did you opt for DreamerV2 instead of DreamerV3?
Q2 Do you have any intuition regarding how the quality of the action-free data affects performance? Since the downstream policy learning relies on behavior cloning, my intuition is that you may need something more intricate than behavior cloning if the data is lower quality or is more diverse.

**Relation To Broader Scientific Literature:**

This paper addresses the important question of extracting knowledge from videos. Natural videos are widely available and contain enormous amounts of information, yet most modern approaches do not tap into that resource. This particular work focuses on videos showing similar tasks but in different environments.

**Theoretical Claims:**

N/A

---

> ### Author Rebuttal · Authors · 2025-04-01
>
> We sincerely appreciate the reviewer's valuable comments and have addressed each point below.
>
> > Q1. The method is compared to DreamerV2 and not V3.
>
> Please refer to our response to **Reviewer LSCr Q1**.
>
> > Q2. This paper doesn't cite FICC, which is another method that tackles a very similar problem.
>
> Thank you for pointing out the FICC paper. We will add the citation and provide a detailed comparison in the revised manuscript. Although FICC and VeoRL both employ VQ-VAE and a forward model to extract latent actions in an unsupervised manner, they are distinct in the following aspects:
>
> **(1) We focus on offline RL.**
>
> While FICC focuses on pre-training a world model with offline data and then transferring it to downstream **online tasks**, our work addresses **offline RL**, where policies must be learned purely from static datasets without online interactions. These are fundamentally distinct problem settings with different technical challenges. For instance, offline RL primarily grapples with distributional shift due to the mismatch between the fixed dataset and real environment dynamics, while FICC employs online fine-tuning to achieve fast adaptation to downstream tasks.
>
> **(2) We handle significant domain gaps.**
>
> FICC operates under the assumption that the source domain (offline data) and target domain (downstream tasks) share the **same** environment. In contrast, our method tackles a more challenging **cross-domain** transfer scenario: extracting knowledge from real-world videos and employing it in a simulation environment. The significant visual and dynamics gaps between real-world and simulated domains substantially increase the difficulty of knowledge transfer.
>
> **(3) We leverage hybrid action spaces for behavioral alignment**
>
> FICC is explicitly designed for **discrete actions** (tested only on Atari) and acknowledges its limitation in handling continuous control tasks ("difficult to handle the continuous action space" as stated in their paper). Our method, however, focuses on the tasks with **continuous actions**. The dual-path policy optimization process we designed associates discrete actions extracted from videos with the continuous action space of the target environment, thereby enhancing offline RL policies with high-level behavioral abstractions (e.g., 'reach', 'push') from video data.
>
> > Q3. Experiments on more difficult, multi-step tasks.
>
> As suggested by the reviewer, we compare our method with DreamerV2 and VIP in **Soccer** from Meta-World. This task typically involves a two-stage decision-making process where the agent must first fetch a ball and then push it to a goal position.
>
> |Soccer|DreamerV2|VIP|Ours|
> |-|-|-|-|
> |Episode return|112.53 $\pm$ 37.90|109.84 $\pm$ 17.32| 231.10 $\pm$ 20.68|
>
> > Q4. The natural videos this method leverages contain, for the most part, high-quality trajectories. This enables the planner to simply use a behavior cloning policy. This however is a quite limiting assumption.
>
> **(1) Do the auxiliary videos contain only high-quality trajectories?**
>
> No, the videos we leverage are **not curated or task-specific**. Instead, they are **diverse, easily accessible, and collected without explicit task filtering**. While these videos may not always align closely with the target tasks (meaning they **should not** be considered high-quality trajectories). Even when the tasks in the videos differ from the target tasks, our method can extract useful priors for policy learning.
>
> **(2) Does the method reduce to simple behavior cloning?**
>
> No, our approach goes beyond naive behavior cloning, as real action labels are unavailable in passive videos. Instead, we learn discrete latent actions that capture abstract behavioral patterns. This allows our method to **generalize across environments**, as it clones high-level behavioral concepts rather than relying on low-level action sequences. By leveraging these latent structures, our method can transfer useful priors even when direct action replication is impossible.
>
> > Q5. 247 right - considering more on -> did you mean "focusing more on"?
>
> Yes.

---

> > ### Comment · Reviewer_W3sg · 2025-04-03
> >
> > Thank you for answering my questions and running additional experiments!
> >
> > Q1: This clarifies your choice of DreamerV2 over V3, thank you.
> >
> > Q2: Thank you for the clarification. I agree that the distinction between FICC and your method is quite big, and I'm not saying you should benchmark against FICC.
> >
> > Q3: Thank you for running this!
> >
> > (1) Do the auxiliary videos contain only high-quality trajectories? and (2) Does the method reduce to simple behavior cloning?
> > I see. The dataset you used for metaworld for example consists of trajectories that are collected using teleoperation. All the trajectories in that dataset are therefore high-quality and solve some kind of task. Granted, most of those tasks may be irrelevant to the task you're testing on. But the high-quality of those trajectories is what enables you to use behavior cloning in the planning network ($\bar a_t = F_\mathrm{BC}(s_t)$). This $\bar a_t$ is then used to condition the policy to execute the high-level behavior $\bar a_t$. How can the high-level behavior cloning policy $F_\mathrm{BC}(s_t)$ provide sensible high-level actions if it isn't trained on any high-quality trajectories relevant to the task at hand? In my understanding, that behavior-cloning policy will perform badly if the offline data has e.g. random trajectories. Is my understanding correct?
> > If so, do you think there could be other ways of training the high-level policy to avoid the limitations of BC?
> >
> > I understand that this is definitely more intricate that naive behavior cloning because you first have to identify the latent actions, and then train your BC policy using those actions, but using BC policy on the high-level has its limitations.

---

> > > ### Author Response · Authors · 2025-04-06
> > >
> > > We would like to thank the reviewer for the prompt reply and ongoing discussion!
> > >
> > > **(1) Clarification on Data Quality and Experimental Setup:**
> > >
> > > Since both the source domain videos and the target domain trajectories are offline in our setup, we would like to briefly restate our experimental setup in the Meta-World environment to avoid any potential misunderstandings:
> > > - Source Domain (**BridgeData Videos**): A large-scale dataset containing diverse robotic manipulation videos, including 50,365 teleoperated demonstrations across 24 environments.
> > > - Target Domain (**Offline Meta-World**): The offline  trajectories used for the target task are **NOT** expert-level, but of medium quality (e.g., suboptimal or unsuccessful).
> > >
> > > **(2) For Low-Quality Source Videos:**
> > >
> > > A key assumption in our work is that the demonstration videos are **large-scale** enough to cover a broader range of skills. This enables us to construct a codebook of latent behaviors that can potentially support cross-domain policy learning. These videos are **readily available** on the Internet. For example, if the target domain is autonomous driving, the vast number of human-driving videos available online can serve as a rich source of demonstrations.
> > >
> > > If the reviewer's concern--"*How can the high-level behavior cloning policy provide sensible high-level actions if it isn't trained on any high-quality trajectories relevant to the task at hand?*"--refers to a scenario where the source data consists only of failed robotic manipulation attempts, or comes from domains that are significantly less related (e.g., using the Something-Something data as the source videos while the target domain is Meta-World), then we acknowledge that the effectiveness of our method in such cases remains uncertain and would require further investigation.
> > >
> > > We agree with the reviewer that using BC at the high level may struggle if the source videos contain entirely random or irrelevant trajectories. If the dataset lacks meaningful high-level behaviors, the extracted latent actions may not provide useful guidance. However, in these cases, since our policy network is learned using a hybrid BC-RL approach, it does not rely solely on BC. The RL component allows the policy to adapt to the target task.
> > >
> > > To partially investigate the impact of using fewer high-quality videos, we have conducted an ablation study analyzing how the quantity of unlabeled source videos affects policy performance. The results are summarized below, indicating that:
> > > - As the number of source domain videos increases, the model's performance improves accordingly, demonstrating that our method effectively leverages information from source domain videos and exhibits strong scalability.
> > > - Even when using only a quarter of the videos, VeoRL's results still significantly outperform DreamerV2 (which does not utilize any source domain videos for training), indicating that our method can effectively extract useful skills from the videos.
> > >
> > > |Handle Press   | All videos  | 1/2 videos | 1/4 videos | DreamerV2 |
> > > | ----- | --- | --- |--- | --- |
> > > |Episode return  |  2651 $\pm$ 620      | 2477 $\pm$ 441      |  1859 $\pm$ 423 |  1201.75 $\pm$ 422.10|
> > >
> > >
> > > Nevertheless, we would like to emphasize that our work focuses on leveraging **action-free** and **cross-domain** demonstration videos, which may involve **different embodied agents and video appearances** from those in the target domain, to facilitate the offline task at hand. We believe this approach broadens the scope of offline RL.
> > >
> > > **(3) For Low-Quality Target Domain Trajectories:**
> > >
> > > As noted in Line 230(left), our framework trains the Plan Net and the corresponding BC model using data from **both** the source domain (BridgeData) and the target domain (Meta-World). This design leverages the diversity of BridgeData to enable the BC model to infer high-level behavioral abstractions (e.g., "reach" or "grasp") that are applicable to the target task (e.g., "Button press"), even when the offline data is of low quality and lacks successful trajectories.
> > >
> > > To validate this, we add a new experiment, in which the offline Meta-World data consists of random trajectories, potentially containing extremely poor demonstrations. The results below indicate that our method could still benefit from the latent skills (estimated by BC) from the auxiliary videos, demonstrating the robustness of our approach in challenging offline RL settings.
> > >
> > > |Button Press (Episode return)|Medium offline data|Random offline data (NEW!)|
> > > |-|-|-|
> > > |Ours|850.20 $\pm$ 169.05|637.85 $\pm$ 150.25|
> > > |DreamerV2|764.87 $\pm$ 120.12|505.47 $\pm$ 150.06|
> > > |VIP| 545.29 $\pm$ 93.67|276.5 $\pm$ 98.8|

---

### Official Review · Reviewer_LSCr · 2025-03-11

**Overall Recommendation:** 2

**Summary:**

This paper introduces a method for offline reinforcement learning through using unlabeled video data to train a world model and doing model based policy optimization. The paper include experimental results from Meta-World, CARLA, and MineDojo.

**Claims And Evidence:**

This paper makes several claims:
1. The proposed method has a performance improvement compared to other offline RL baselines.

This claim is generally supported with evidence from Meta-World, CARLA, and MineDojo environments. However, the paper states that the DreamerV2 algorithm is a main baseline without acknowledging DreamerV3, which is a newer method in the same area.

2. Better offline to online transfer.

While it does show this compared to DreamerV2, it could use more empirical evidence comparing to other methods.

3. Overestimation bias.

The claim is made that overestimation bias is the key challenge in offline reinforcement learning, but does not show if the proposed method mitigates overestimation in the value function.

**Essential References Not Discussed:**

The key contribution is a method for model based offline RL from unlabeled videos. It does not compare with DreamerV3, which is a model based RL algorithm that builds on DreamerV2, a main baseline in this paper.

**Experimental Designs Or Analyses:**

The experimental design is generally sound. The paper compares VeoRL against both model-based methods such as DreamerV2 and LOMPO and model-free methods such as CQL.

**Methods And Evaluation Criteria:**

The benchmark datasets of Meta-World, CARLA, and MineDojo make sense for the problem or application at hand.

**Other Comments Or Suggestions:**

I do not have other suggestions.

**Other Strengths And Weaknesses:**

The paper is original and significant. The main weakness is it does not compare with an important baseline which is DreamerV3.

**Questions For Authors:**

1. Value overestimation: in the paper it is claimed overestimation bias the key issue in offline RL, but don't provide measurements for value function estimates in the proposed method. Did you compare estimated vs actual values in the experiments?

2. Baseline comparisons: the main baseline for the paper is DreamerV2. Is there a reason DreamerV3 is not compared to in the paper?

**Relation To Broader Scientific Literature:**

The key contributions of the paper build on several important research threads in offline reinforcement learning namely leveraging unlabeled videos for RL and model based offline RL.

**Theoretical Claims:**

There are no substantial theoretical claims made in the paper.

---

> ### Author Rebuttal · Authors · 2025-04-01
>
> We appreciate the reviewer’s insightful comments and provide our responses below.
>
> > Q1. Comparison with DreamerV3.
>
> The decision to use DreamerV2 as the backbone stems from two key considerations:
> - **Offline setting performance:** While DreamerV3 achieves strong performance across diverse tasks with fixed hyperparameters, our experiments (as shown below) demonstrate that it underperforms in offline RL settings without hyperparameter tuning compared to DreamerV2.
> - **Backbone consistency:** LAMPO, a key baseline in our work, employs DreamerV2 as its backbone. To enable direct and equitable comparisons between our method and LAMPO, we maintained consistency by using the same backbone.
>
> Notably, we have conducted additional experiments on Meta-world by integrating VeoRL with DreamerV3. The results show that our approach outperforms vanilla DreamerV3, showing its ability to seamlessly integrate with different model architectures.
>
> |Episode return|DreamerV2|DreamerV3|VeoRL(DV2)|VeoRL(DV3)|
> |-|-|-|-|-|
> |Drawer Open|1168.35 $\pm$ 59.55|674.55 $\pm$ 79.04|1953.6 $\pm$ 121.48|1393.5 $\pm$ 122.5|
> |Handle Press|1201.75 $\pm$ 422.10|257.85 $\pm$ 247.05|2650.90 $\pm$ 619.60|1360.15 $\pm$ 547.85|
>
> |Success rate|DreamerV2|DreamerV3|VeoRL(DV2)|VeoRL(DV3)|
> |-|-|-|-|-|
> |Drawer Open|0.18 $\pm$ 0.04|0.00 $\pm$ 0.00|0.70 $\pm$ 0.07|0.55 $\pm$ 0.15|
> |Handle Press|0.33 $\pm$ 0.11|0.05 $\pm$ 0.05|0.60 $\pm$ 0.12|0.35 $\pm$ 0.15|
>
> > Q2. Offline to online transfer.
>
> In the offline-to-online transfer setup, we used DreamerV2 as the comparison model, as it has demonstrated stable performance across multiple environments. In this rebuttal, we conduct further experiments with the second-best comparison models in each environment (VIP, LOMPO, and VPT, respectively). The results presented below consistently demonstrate that our approach outperforms existing methods. The updated Fig 6 is available at https://anonymous.4open.science/r/image-765.
>
> |Meta-World return (Soccer)|20K|60K|100K|
> |-|-|-|-|
> |DreamerV2|37.22|81.41|106.92|
> |VIP|18.52|29.03|117.31|
> |VeoRL|150.09|213.34|234.80|
>
> |CARLA return (Night)|20K|60K|100K|
> |-|-|-|-|
> |DreamerV2|2.01|15.17|42.95|
> |LOMPO|−20.45 |-12.05|-1.26|
> |VeoRL|27.24|37.61|79.07|
>
> |MineDojo success rate (Cobblestone)|50K|100K|150K|
> |-|-|-|-|
> |DreamerV2|0.27|0.41|0.70|
> |VPT|0|0|0|
> |VeoRL|0.58|0.86|1.00|
>
> > Q3. Value overestimation.
>
> **(1) Does VeoRL alleviate value overestimation?**
>
> Overestimation bias is indeed a key challenge in offline RL, and our method mitigates this indirectly through intrinsic rewards derived from auxiliary videos. To further validate its impact, we compare estimated values with true values (i.e., the discounted sum of actual returns) on the Meta-World Handle Press task.
> - We analyze the distribution of estimated and true values using histograms. The results below show that VeoRL's predicted values align more closely with the true values, whereas DreamerV2 exhibits significant overestimation. Specifically, in 69.12% of states, DreamerV2 predicts values exceeding 1000, despite 57.71% of true values being below 200.
>
> Notably,
> - Since our model incorporates both environmental and intrinsic rewards in value estimation, we also include the intrinsic reward when computing the true value. The negative intrinsic reward leads to a lower overall highest true value in VeoRL. As a result, directly comparing the true values of the two models is not meaningful.
> - Additionally, value computation follows each model’s successful trajectories.
>
> Corresponding figures are available at https://anonymous.4open.science/r/image-765. Our approach is shown to produce value estimates that are closer to the true values compared to DreamerV2, indicating reduced overestimation bias.
>
> |Histogram of 50 episodes|[0,200)|[200,400)|[400,600)|[600,800)|[800,1000)|[1000,1200]|
> |-|-|-|-|-|-|-|
> |DreamerV2 estimate|0.44%|3.12%|5.20%|6.45%|15.68%|69.12%|
> |True value|57.71%|5.08%|5.73%|6.34%|25.14%|0%|
>
> |Histogram of 50 episodes|[0,200)|[200,400)|[400,600)|[600,800)|[800,1000)|[1000,1200]|
> |-|-|-|-|-|-|-|
> |VeoRL estimate|1.98%|11.79%|30.69%|55.54%|0%|0%|
> |True value|15.93%|24.51%|49.98%|9.58%|0%|0%|
>
> **(2) How does VeoRL specifically handle the value estimation bias?**
>
> VeoRL leverages **intrinsic rewards** to mitigate value bias in offline RL by **grounding value estimates in realistic behaviors**:
> - Traditional value functions in offline RL can extrapolate poorly for unseen states because they optimize based only on limited environmental feedback.
> - Intrinsic rewards encourage high-value states to align with realistic source behaviors. By incorporating intrinsic rewards from video priors, our approach provides an alternative supervision signal, reducing reliance on extrapolated Q-values.
> - This approach acts as a regularization mechanism, constraining the model from assigning arbitrarily high values to unfamiliar states, preventing unrealistic value spikes.

---

### Official Review · Reviewer_e9tP · 2025-03-12

**Overall Recommendation:** 3

**Summary:**

This paper presents a model-based RL method, Video-Enhanced Offline RL (VeoRL), which leverages unlabeled Internet videos to enrich the world model. The world model comprises two state transition branches: through real actions using a trunk net and through latent behaviors using a plan net. And the policy is trained by two-stream imagined trajectories generated by the trunk net and the plan net.

**Claims And Evidence:**

Yes.

**Essential References Not Discussed:**

None.

**Experimental Designs Or Analyses:**

Yes.

**Methods And Evaluation Criteria:**

Yes.

**Other Comments Or Suggestions:**

1. The paper could include a quantitative analysis of computational complexity in the experimental section, particularly in comparison with other offline RL methods.

2. In the experimental section, the authors use only 3 seeds. However, I believe that this is not enough. And I recommend the use of more than five seeds to ensure robustness.

**Other Strengths And Weaknesses:**

**Strength**：

1. The paper is well-written and easy to follow.

2. The method leverages unlabeled internet video data to enhance offline RL performance and addresses cross-domain distribution shifts.

3. The paper provides extensive experimental validation across multiple visuomotor control tasks (e.g., robotic manipulation, autonomous driving, and open-world games), demonstrating that VeoRL significantly outperforms existing offline RL methods on multiple tasks.

**Weaknesses**:
1. **Computational Complexity**: The paper mentions that VeoRL's training process involves extracting latent behaviors and optimizing a dual-branch world model, which may lead to high computational complexity and memory requirements. However, the paper does not provide a detailed quantitative analysis of the computational overhead or compare it with other methods.

2. **Interpretability of Latent Behaviors**: While the paper shows the correspondence between latent behaviors and real actions, the semantic interpretation of these latent behaviors remains unclear. For example, the paper does not discuss in detail how these latent behaviors specifically guide policy optimization or their generalizability across different tasks.

**Questions For Authors:**

The paper uses datasets like BridgeData-V2 and NuScenes as auxiliary video sources but does not provide an in-depth analysis of their diversity and suitability. Did the authors perform an analysis of the distribution differences between these datasets and the target tasks? If so, could more details be provided in the revised version?

**Relation To Broader Scientific Literature:**

This contribution is closely related to existing offline RL methods (e.g., CQL, DreamerV2) and video-enhanced RL approaches (e.g., APV, VIP).

**Theoretical Claims:**

The paper does not contain any proofs or theoretical developments; hence, it is unnecessary to perform a check.

---

> ### Author Rebuttal · Authors · 2025-03-31
>
> We thank the reviewer for the constructive comments.
>
> > Q1. Complete code documentation and environment configuration files.
>
> Code and full configuration files will be open-sourced upon acceptance.
>
> > Q2. Computational complexity.
>
> Below, we present the training time and GPU usage until model convergence. While the latent behavior extraction and dual-branch architecture in VeoRL introduce additional computations, our experiments demonstrate a favorable efficiency-performance trade-off:
> - **Compared with DreamerV2:** VeoRL requires 1.3× GPU memory and 2× training time. This modest cost results in about 360% improved episode return in CARLA and 140% higher success rates in MineDojo, as shown in Fig 1.
> - **Compared with APV, VIP, and LOMPO:** VeoRL shows comparable training time and offers advantages in lower memory usage.
>
> ||VeoRL|DreamerV2|APV|VIP|LOMPO|
> |-|-|-|-|-|-|
> |Training time|19h|9h|19h|20h|25h|
> |GPU memory|5126M|3892M|6039M|12292M|8841M|
>
> > Q3. Interpretability of latent behaviors.
>
> **(1) Semantic interpretation of latent behaviors:**
>
> Latent behaviors align with real actions through our model, guided by the reward function and optimization objectives. As visualized in Fig 7, for example, in BridgeData, distinct latent behaviors correspond to atomic actions like "grasp", "reach", or "push".
>
> **(2) How do the latent behaviors guide policy optimization?**
>
> Latent behaviors are abstract and high-level, requiring alignment with low-level actions in the target task. To enable latent behaviors to guide target policies, we modify the MBRL algorithm as follows:
> - The policy network conditions on both the current state and latent actions from the behavior cloning module, enabling high-level intentions to guide low-level actions.
> - State transitions are modeled using a Trunk Net (driven by real low-level actions) and a Plan Net (guided by high-level latent behaviors) for long-horizon planning.
> - An intrinsic reward encourages the rollouts from the Trunk Net to progressively align with long-term states predicted by the Plan Net, bridging the target policy with behaviors extracted from auxiliary videos.
>
> **(3) Generalizability across tasks:**
>
> We conduct additional experiments on Meta-World and MineDojo to validate the transferability of latent behaviors across tasks. The results are presented below.
>
> Training schemes: The behavior abstraction network is frozen after training on Task A, with a fixed latent codebook. For Task B, this pre-trained network is directly deployed without undergoing task-specific fine-tuning on source video data.
>
> The minimal performance degradation confirms that our method's latent behaviors trained on Task A remain effective for Task B, consistently outperforming other baselines trained specifically on Task B. This transferability significantly reduces training costs for downstream tasks, as it spares the time for re-training on the auxiliary videos.
>
> |Meta-World|Codebook construction|Downstream task|Success rate|
> |-|-|-|-|
> |VeoRL|Button Press|Button Press|0.62 $\pm$ 0.02|
> |VeoRL|Handle Press|Button Press|0.63 $\pm$ 0.12|
> |DreamerV2|NA|Button Press|0.58 $\pm$ 0.08|
> |VIP|NA|Button Press|0.18 $\pm$ 0.08|
>
> |MineDojo|Codebook construction|Downstream task|Success rate|
> |-|-|-|-|
> |VeoRL|Harvest sand|Harvest sand|0.55 $\pm$ 0.05|
> |VeoRL|Harvest water with bucket|Harvest sand|0.50 $\pm$ 0.10|
> |DreamerV2|N/A|Harvest sand|0.25 $\pm$ 0.06|
> |VPT|NA|Harvest sand|0.20 $\pm$ 0.04|
>
> > Q4. More seeds.
>
> As suggested, we have conducted additional experiments on MineDojo using five different random seeds. The updated results below show consistent performance of our model. We'll include full results in the revised paper.
>
> ||VeoRL|DreamerV2|LOMPO|VPT|
> |-|-|-|-|-|
> |Harvest log in plains|0.40 $\pm$ 0.07|0.13 $\pm$ 0.08|0.03 $\pm$ 0.04|0.02 $\pm$ 0.01|
> |Harvest sand|0.55 $\pm$ 0.05|0.25 $\pm$ 0.06|0.05 $\pm$ 0.05|0.20 $\pm$ 0.04|
>
> > Q5. The distribution differences between video data and the target tasks.
>
> We summarize the differences between the source and target domains below. These distinctions highlight VeoRL’s ability to effectively leverage real-world video data, even with notable distributional discrepancies, to enhance target domain task performance.
>
> ||Source: BridgeData-V2 | Target: Meta-World|
> |-|-|-|
> |Robot arm|WidowX 250 6DOF|Simulated Sawyer|
> |Data source|Collected from real robots|Generated in simulated environments|
> |Task design|Real-world tasks (e.g., kitchen activities, tool use)| Standardized tasks (50 predefined tasks)|
> |Camera view|Random-view|Right-view|
> |Episode length|Random|500|
>
> ||Source: NuScenes| Target: CARLA|
> |-|-|-|
> |Data source|Real-world data collected in Boston and Singapore using sensor-equipped vehicles|Synthetic data generated via an open-source simulation platform|
> |Scenarios|1000 real-world scenarios (e.g., lane change, unprotected turn, jaywalker)|Customizable scenarios (e.g., dynamic weather, traffic density)|
> |Camera number|6|1|
> |Episode length|Random|1000|

---

> > ### Comment · Reviewer_e9tP · 2025-04-09
> >
> > Thank the authors for a detailed response to my review.
> >
> > The authors provided quantitative comparisons of training time and GPU usage to clarify the computational overhead. The addition of experiments with five random seeds for MineDojo strengthens the statistical reliability of the results, mitigating concerns about variance.
> >
> > The rebuttal lists domain differences (e.g., real vs. simulated robots) but does not explain how VeoRL mitigates these shifts. Ablation studies on the MMD loss or other domain adaptation components would clarify their contribution.

---

### Decision · Program_Chairs · 2025-05-01

**Decision:**

Accept (poster)

**Comment:**

This paper is a borderline case, and as such I conducted a short review myself.

The paper proposes an approach to leverage unlabelled video data for offline RL with a world model. It connects the two modalities by training a latent action model from unlabelled videos and a regular world model from labelled, and then training the agent across both. The rebuttal addressed several concerns, notably 1) comparison with DreamerV3, 2) computational cost 3) ablation on the quantity of unlabelled video.

Overall I believe this could be an interesting contribution to ICML 2025. My main issue with the work is the lack of relation to the Genie paper, which won a Best Paper award at ICML 2024. I think the authors could *strengthen* the impact of their work by making this connection clear. For instance, Genie showed the power of pre-training with latent actions at scale, but did not provide an explicit path to using this for transferring policies to real environments. This paper could position itself as one path for this, potentially unlocking large-scale video world models.